# The Cerebellar Involvement in Autism Spectrum Disorders: From the Social Brain to Mouse Models

**DOI:** 10.3390/ijms23073894

**Published:** 2022-03-31

**Authors:** Lisa Mapelli, Teresa Soda, Egidio D’Angelo, Francesca Prestori

**Affiliations:** 1Department of Brain and Behavioral Sciences, University of Pavia, 27100 Pavia, Italy; teresa.soda01@universitadipavia.it (T.S.); dangelo@unipv.it (E.D.); 2Brain Connectivity Center, IRCCS Mondino Foundation, 27100 Pavia, Italy

**Keywords:** cerebellum, autism spectrum disorders, excitatory/inhibitory balance, mouse models of ASD, social brain

## Abstract

Autism spectrum disorders (ASD) are pervasive neurodevelopmental disorders that include a variety of forms and clinical phenotypes. This heterogeneity complicates the clinical and experimental approaches to ASD etiology and pathophysiology. To date, a unifying theory of these diseases is still missing. Nevertheless, the intense work of researchers and clinicians in the last decades has identified some ASD hallmarks and the primary brain areas involved. Not surprisingly, the areas that are part of the so-called “social brain”, and those strictly connected to them, were found to be crucial, such as the prefrontal cortex, amygdala, hippocampus, limbic system, and dopaminergic pathways. With the recent acknowledgment of the cerebellar contribution to cognitive functions and the social brain, its involvement in ASD has become unmistakable, though its extent is still to be elucidated. In most cases, significant advances were made possible by recent technological developments in structural/functional assessment of the human brain and by using mouse models of ASD. Mouse models are an invaluable tool to get insights into the molecular and cellular counterparts of the disease, acting on the specific genetic background generating ASD-like phenotype. Given the multifaceted nature of ASD and related studies, it is often difficult to navigate the literature and limit the huge content to specific questions. This review fulfills the need for an organized, clear, and state-of-the-art perspective on cerebellar involvement in ASD, from its connections to the social brain areas (which are the primary sites of ASD impairments) to the use of monogenic mouse models.

## 1. Introduction

Autism spectrum disorders (ASD) are complex neurodevelopmental disabilities characterized by impairments in social cognition in the presence of restricted, repetitive, routinized behaviors, interests, or activities (i.e., stereotyped and repetitive speech, movements, and inflexible adherence to routines). This definition is compatible with the diagnostic criteria for ASD defined in the Diagnostic and Statistical Manual of Mental Disorders, 5th Edition [1]. Social cognition refers to specific mental abilities that enable us to interpret, analyze, store, and apply information appropriately about the social environment [2,3]. Deficits in joint attention, emotion recognition, social perception, and verbal and nonverbal communication (i.e., body language and facial expression processing) have been identified as core cognitive deficits in ASD [1]. They also include the ability to understand other people’s goals, intentions, and mental states, known as empathy, mentalization or theory of mind [4,5,6]. Moreover, a range of comorbid disorders frequently accompany individuals with ASD, including psychiatric illness (i.e., bipolar disorder, schizophrenia, obsessive-compulsive disorder), epilepsy, sleep disruption, hyperactivity, and gastrointestinal symptoms [7,8]. Moreover, immune profile alterations during early life may contribute to neurodevelopment disorders including ASD [9,10,11,12,13]. Indeed, the presence of persistent neuroinflammation in postmortem brain tissue from ASD individuals is a prominent feature [14]. For an extensive review on neuroinflammation in ASD see [10,11,12,13]. Although familial and monozygotic twins studies have shown heritability estimated at 60–90%, indicating a high genetic contribution [15,16,17], the precise cause of ASD remains largely elusive, despite recent advances that have recognized the roles of various forms of genetic variants (i.e., common vs. rare; autosomal inherited vs. *de novo*; dominant vs. recessive) [18,19,20]. Common genetic variants with small effect sizes have been mainly evaluated to explain the ASD liability while rare or de novo mutations, representing a single event with considerable effect, are responsible for <5–20% of the cases [21,22]. In addition to genetic factors, several environmental risk factors have been associated with ASD, including advanced parental age, fetal environment (i.e., maternal inflammation and diseases), fertility treatments, medication (valproate, selective serotonin reuptake inhibitors), nutritional factors (i.e., iron, zinc, and copper), chemicals and toxicants (i.e., air pollution, pesticides) [23,24]. Although there is wide acceptance that ASD has multiple genetic and environmental causes, a complete understanding of how they interact to impact ASD etiology is still missing, making the investigation of its pathophysiology challenging. The advent of high-throughput sequencing represented the most successful aspect of ASD gene research to date, leading to identification of monogenic disorders associated with ASD, which, although individually rare, are estimated to represent approximately 10% of all ASD cases [25,26]. Additionally, monogenic mouse models showing ASD phenotype are easily generated through selective genetic manipulations and are essential to moving the research field forward. Advances in research techniques for the study of brain–behavior relationships in animals and humans have allowed elucidation of the neural basis of ASD by linking specific social cognition deficits to dysfunction in specific brain structures and circuits, accordingly denominating “the social brain” [27,28,29]. Within these circuits, the cerebellum, traditionally considered to be almost exclusively involved in motor learning and coordination, has been recently recognized to contribute to cognitive processing and social behavior [30,31,32]. Herein, we will first describe the social brain network that is primarily impaired in ASD. Though social brain areas and functionally connected regions are at the core of ASD pathophysiology, it is beyond the scope of this review to comprehensively summarize the anatomical and functional alterations found in these regions. Therefore, we will then present a broad range of research proving that the cerebellum is essentially involved in ASD. Lastly, we will summarize the main findings from monogenic mouse models of ASD, in which the pathophysiology of cerebellar dysfunction has been documented. We believe that increased attention to the role of cerebellar pathology in ASD etiology and ASD-related behaviors will provide new insights into the pathogenetic mechanisms that can generate novel molecular/cellular/anatomic-targeted therapeutics.

## 2. Neural Bases for Impaired Social Cognition in ASD

In humans, focal brain lesions and social task-based fMRI studies have largely contributed to identifying a network of brain regions (called “the social brain”) implicated in social cognition [33,34,35,36,37]. The primary regions of the social brain include the medial prefrontal cortex (mPFC) [38,39], the temporoparietal junction (TPJ), the posterior superior temporal sulcus (pSTS), the inferior frontal gyrus, the anterior cingulate cortex (ACC) [40], and the anterior insula (AI) [41] (Figure 1). Moreover, the hippocampal formation, the ventral tegmental area (VTA), the nucleus accumbens (NAcc), the amygdala, and the cerebellum are highly connected with the social network structure, acting as important functional hubs [42,43,44,45,46,47,48]. In ASD subjects, several studies identified a combination of atypical structural and functional features in these areas. Structurally, cortical and subcortical measurements in ASD postmortem brain tissue, primarily in frontal and temporal cortices and the amygdala, described an aberrant organization, such as small cell size and increased packing density [49,50], white matter volume increase [51], decreased cortical thickness [52], and more numerous and narrower minicolumns [53]. Functionally, a growing number of studies used fMRI to examine changes in intrinsic functional connectivity (FC) of specific brain regions and circuits [54,55] between individuals with ASD and normally developing controls. In most cases, FC analysis indicated that ASD subjects exhibit long-range under-connectivity and local over-connectivity [56,57,58,59,60,61,62,63]. Long-range under-connectivity between PFC and posterior brain regions were most often reported [56,57,58,64,65], but it was also described in other regions, as between the amygdala and temporal cortex [66], the supplementary motor areas and the thalamus [67], the PFC and premotor and somatosensory cortices [68], and among the PFC, amygdala, and hippocampus [69]. However, some studies reported increased FC among individuals with ASD [69]. Specifically, long-range over-connectivity was discovered within thalamocortical [70], striatocortical [71], and corticocortical circuits [72]. In contrast, local over-connectivity in ASD is less solidly determined. For instance, local over-connectivity was found in ASD in the extrastriate cortex, frontal and temporal regions, amygdala, and parahippocampal gyri [73,74,75,76,77]. Other studies, however, reported a reduction of local connectivity, principally in the fusiform face area and in the somatosensory cortex [78,79], or a combination of both patterns [80]. Several experimental evidence suggested a higher excitatory/inhibitory (E/I) ratio as a possible correlate for local over-connectivity [81,82,83], for example, through an increased glutamatergic or reduced GABAergic signaling [84,85]. Cortical GABAergic neurons are thought to control the functional integrity and segregation of minicolumns via lateral inhibition [86]. Casanova and colleagues [53] found significant differences between frontal and temporal cortices of ASD and typically developing individuals in the number of minicolumns, in the horizontal space between minicolumns, and their internal structure. Ultimately, minicolumns were more numerous, smaller, and less compact in their cellular configuration. Mechanisms underlying this deficit are still unknown. Moreover, GABAergic neurotransmission is involved in generating gamma-band oscillatory activity [87]. Gamma-band oscillations are involved in a wide range of cognitive processes from the perception of gestalt [88] to selective attention [89,90,91] and working memory [92,93]. Magnetoencephalography (MEG) and electroencephalography (EEG) studies have reported correlations between gamma-band oscillatory activity and ASD severity as measured by the Social Responsiveness Scale [94,95,96]. It should also be considered that an unbalanced E/I ratio might be amplified by delayed brain development, resulting in retardation of synaptogenesis, pruning, and myelination [97,98,99]. Lastly, strong evidence is also reported for alterations in glutamatergic signaling pathways in ASD, involving metabotropic glutamate receptor 5 (mGluR5) upregulation and genetic aberrations associated with NMDA receptors [100,101]. However, the scenario is much more complicated, with both increases and decreases in glutamate-mediated signaling reported in association with the ASD phenotype [102]. Overall, the above-presented data lend support to models hypothesizing well-defined neural substrates of social cognition and propose specific neural bases that may govern social cognitive impairments in ASD. By contrast, further investigations are needed to better understand the complex interactions between social brain areas, connectivity, frequency bands, and physiological aspects (i.e., roles of specific cell types, maturational processes, receptors) and how they relate to different cognitive processes.

## 3. Cerebellar Involvement in ASD

### 3.1. Cerebellar Circuit Microanatomy

The cerebellum, meaning “little brain” in Latin, has been historically considered a subcortical motor structure that controls the coordination of voluntary movements, balance, posture, and muscle tone. Furthermore, it contributes to different forms of motor learning. There is now robust evidence that the cerebellum may be related to a variety of cognitive and emotional functions such as language, attention, fear, and pleasure responses [30,32,103,104]. The cerebellum is composed of tightly folded layers of grey matter forming the cerebellar cortex, with the white matter underneath surrounding four deep cerebellar nuclei (DCN) [105]. The cerebellar cortex is organized into three layers. The outer molecular layer (ML) is composed of two types of inhibitory neurons: stellate (SCs) and basket cells (BCs). The Purkinje cell layer consists of a large pear-shaped Purkinje cell (PC) soma monolayer. The inner granular layer is composed of excitatory granule cells (GrCs) and inhibitory Golgi cells (GoCs). The primary input pathways entering the cerebellum are the mossy fibers (MFs) and climbing fibers (CFs). In the granular layer, MFs directly synapse on the dendrites of GrCs, whose axons ascend toward the ML, where they bifurcate to form T-shaped branches named parallel fibers (PFs) [106,107,108]. PCs receive excitatory input from PFs and CFs, which originate in the inferior olive (IO) [109] and project their axons to DCN neurons. DCN neurons provide the final output of the cerebellum by integrating inhibitory and excitatory inputs from PC axons, MF, and CF collaterals, respectively [110,111]. The activity of PCs is modulated by three types of inhibitory interneurons that are activated by PFs and classified into two main types: BCs and SCs, which are found in the ML, and GoCs, located in the granular layer. Specifically, BCs are found in the deep ML and their axons form pericellular nests in close proximity to PC soma as well as specialized terminals known as pinceaux surrounding the initial segment of PC axons. SCs are located in the upper ML and their axons terminate on the shafts of PC dendrites [112,113]. GoCs receive excitatory synaptic input from MFs on the basal dendrites and PFs on the apical dendrites [114,115], and their axons make inhibitory synapses with GrCs [116,117]. Thus, GoC activity indirectly affects PC output by modulating GrC discharge [112,118] (Figure 2).

### 3.2. Cerebellar Connectivity to Social Brain Areas

Experiments using task-based fMRI and positron emission tomography (PET) revealed that separate regions of the cerebellum are associated with distinct cerebral areas through polysynaptic circuits, forming a functional topography [119,120,121,122]. The sensorimotor cerebellum is represented in the anterior lobe (lobules I-V) and lobule VIII, while the cognitive cerebellum comprises the posterior lobe (lobules VI and VII), including hemispheric extensions (CrusI/CrusII) [122]. Finally, the posterior vermis and hemispheres represent the limbic cerebellum [122,123,124]. The DCN send direct projections to the ventrolateral (VL) and the intralaminar thalamic nuclei, particularly the dorsomedial (MD) nucleus [125]. The VL nucleus, classically known as an integrative center for sensorimotor transformations, targets the primary motor cortex (M1) [42,126,127], whereas the MD nucleus, like other intralaminar nuclei, has widespread cortical projections including the mPFC (Figure 3A) and the superior temporal sulcus [128,129,130,131,132]. Past anatomical studies demonstrated that the cerebellum is interconnected with parts of the limbic system, including the hippocampus, amygdala, and cingulate cortex (Figure 3B) [133,134]. Recently, Bohne and colleagues [135] reported an elegant tracing study identifying a new cerebellar-hippocampal connection via the VL thalamic nucleus in mice. In support of this finding, unilateral removal of the cerebellar hemispheres [136] or PC signaling deficits [137] determined an impairment in hippocampal-based behavioral tasks as goal-directed navigation tests. Furthermore, monosynaptic projections originate in the hippocampus to primarily target the PFC in rodents and primates [138,139]. Several shreds of evidence obtained using different methodologies show that the cerebellum and amygdala are connected [140]. For example, Sang and colleagues found functional connectivity between cerebellar lobules I-V and the amygdala, analyzing resting-state fMRI in healthy young adults [141]. Heath and Harper, recording evoked potentials or using histological tract-tracing, showed connections between DCN and amygdala in cats and monkeys [142]. Finally, Morris and colleagues found amygdala and cerebellum coactivation during the presentation of facial expressions in human subjects [143]. The cerebellum is also connected with the cingulate cortex indicating its involvement in motivational and emotional processing [144]. Early animal studies showed electrophysiological responses in the ACC following electrical stimulation of the vermis area [142,145]. These results were confirmed almost forty years later by Krienen and Buckner using resting-state fMRI in young adults, showing that CrusI and anterior cingulate cortex were functionally connected [146]. Lastly, electrical stimulation of DCN was reported to evoke dopamine release in the mPFC in rodents [147,148,149]. Cerebellar modulation of dopamine release onto the mPFC could be mediated by two separate neuronal pathways originating from the DCN. The first one activates the mesocortical dopaminergic pathway via reticulo-tegmental nuclei (RTN), which, in turn, project to pedunculopontine nuclei (PPT) and then directly stimulate VTA dopaminergic cells that send their axons to the mPFC [133,150,151,152,153]. The second is by modulation of mesocortical dopaminergic release via glutamatergic afferents originating in the thalamic nuclei (VL and MD) [43,154,155] (Figure 3B). More intriguingly, a recent study has shown, using optogenetic manipulation, the existence of a direct cerebellum-VTA pathway suggesting a prominent role of the cerebellum in modulating social behavior [156]. The primary cerebellar connections reported above are summarized in Figure 3. Altogether, these findings propose that dysfunctions described within the cerebral cortical network, usually associated with social features of ASD, could be at least partly related to an impaired connectivity between the cerebellum and key social brain areas.

### 3.3. Cerebellar Structural Abnormalities in ASD

The cerebellum is the brain structure most constantly found abnormal in ASD, and an increased risk for ASD is dependently associated with cerebellar damage [157,158,159,160,161]. Early anatomical studies examining postmortem ASD brain tissue reported a significant reduction in the number of PCs in the lateral hemisphere compared with the medial vermis [162,163]. In subsequent years, the reduction in PC density has been widely documented (about 75% of ASD cases reported in the literature [164,165,166,167]). Fatemi and colleagues [168] found a reduction in PC size in about 25% of ASD cases. Additionally, a variable decrease in GrCs numerosity was reported [169], while the molecular interneurons were preserved [170]. Animal models of spontaneous cerebellar mutations are frequently characterized by PC loss, often showing a failure in the regression of multiple innervations of PCs by CFs, with each PC receiving up to four CF inputs [171,172,173] instead of a single one [174]. A similar PC hyper-innervation due to decreased PC number has not yet been described in human ASD brains, but it could provide support to IO neurons, which are unaffected in number [175,176,177]. Concerning the DCN, the neurons presented an enlarged size at a younger age, while older ASD cerebella showed abnormally smaller neurons, also reduced in number in fastigial and interposed nuclei [169,176]. MRI has emerged as a powerful tool for visualization and diagnostics of neuroanatomical abnormalities in ASD since its inception in the 1980s. Nevertheless, many results are contradictory due to the heterogeneity of underlying causes and the complexity of this disorder. Structural MRI studies in ASD patients described a reduction in the size of several regions of the cerebellum including the vermis, particularly the lobules VI and VII [176,178,179,180,181,182]. In contrast, Piven and colleagues [183] did not confirm these findings but revealed an enlarged cerebellar volume in ASD patients. Interestingly, the cerebellar volume was proportionally scaled to the total brain volume [184]. Further findings included an enlargement in cerebellar white matter volume and a reduction in the grey/white matter ratio [185,186]. In conclusion, the abnormalities of cerebellar structural integrity may be considered as significant predictive factors of ASD and cause differences in functional connectivity reported in ASD young adults (see paragraph below).

### 3.4. Cerebellar Functional Abnormalities in ASD

Studies in children and young adults with ASD, using resting-state FC (rsFC) techniques, documented a general cerebro-cerebellar over-connectivity [187]. However, both under- and over- connectivity have been observed depending on the brain regions investigated. For example, rsFC were increased between non-motor areas of the cerebellum (lobules VI and CrusI) and sensorimotor cerebral cortical regions, such as premotor and primary motor cortices, somatosensory temporal cortex, and occipital lobe; and decreased in cerebro-cerebellar circuits involving language and social interaction, particularly between CrusI/II and PFC, posterior parietal cortex, and the inferior/middle temporal gyrus [157,187,188]. It should be noted that no specific correlation between FC and behavioral profiles in individuals with ASD has been established [74,189,190,191], although novel findings reported abnormalities in FC related to ASD symptom severity [192,193]. To date, very few studies addressed the FC between the cerebellum and cortical regions, focusing on motor task performance in ASD. For example, during self-paced sequential finger tapping, fMRI in children with ASD did not display the activation in the lobules IV/V and in the anterior cerebellum present in typically developing groups [67]. Furthermore, Jack and Morris [194] investigated coordinated activity between the neocerebellum (particularly CrusI) and pSTS during a task that requires perception and use of information about others’, and remarkably found stronger CrusI–pSTS connectivity positively associated with mentalizing ability, in young adults with ASD. Therefore, together with the structural data described above, these findings are consistent with the idea that ASD is a disorder characterized by abnormalities in cerebellum-cerebral functional connectivity, which could be related to symptom severity.

### 3.5. Cerebellar Neurochemical Abnormalities in ASD

Neurochemical research has progressed in the last 20 years and has produced promising results. For example, reelin expression was reduced in the cerebellum of ASD individuals [195]. This glycoprotein regulates proper cortex lamination and neuronal migration during development and adult life, sustaining cell signaling and synaptic function. Furthermore, serotonin concentration is also altered in the ASD cerebellum. Specifically, Chugani and colleagues [196,197], using PET scanning with a tracer for serotonin synthesis in ASD young adults, reported reduced serotonin levels in the thalamus and the frontal cortex associated with increased serotonin concentration in DCN. Serotonin is well known for its role in neurodevelopment, regulating cell migration and proliferation [198], neurite outgrowth, and neuronal survival [199] as well as synaptogenesis [200]. Therefore, aberrant serotoninergic neuromodulation of dentatothalamocortical pathway connecting the cerebellum with social structures could compromise cognitive and behavioral maturation in ASD. Additionally, a reduced expression in PCs of glutamic acid decarboxylase 67 (GAD67) mRNA, an essential enzyme for converting glutamate to GABA, is a consistent finding in the postmortem cerebellum of ASD patients [201,202]. Conversely, a higher expression of GAD67 mRNA in cerebellar molecular layer interneurons was observed [203], suggesting the existence of an upregulation mechanism to counterbalance the altered inhibition of DCN by PCs. Interestingly, the larger-sized subpopulation of GABAergic neurons in the DCN, which project specifically to the IO [126,204], were reported to exhibit a reduction in GAD65 mRNA expression [205]. Thus, GABAergic neurotransmission alteration in DCN could profoundly affect olivary oscillations and subsequently affect the timing of PC activity (Figure 4). To date, accumulating evidence hints for the hypothesis that core features of ASD emerge from disturbances in the E/I balance within neural circuits [81,85,175,206]. In conclusion, the above findings highlight the role of reelin, serotonin concentration, GABAergic neurotransmission and GAD enzymes in ASD. However, more investigations are needed to better evaluate the mechanisms underlying E/I balance.

### 3.6. Cerebellar Inflammation in ASD

Cells of the immune system and their products are able to directly regulate neuronal function, cell migration, proliferation, adhesion, differentiation [207], and synapse formation and plasticity [208]. Thus, they play crucial roles in modulating neuronal circuits that constitute the basis for human social cognition and behavior [160]. Immune profile alterations have been described in ASD patients [9,10,11,12,13] and may contribute to the disorders. Postmortem brain tissue from ASD individuals often shows persistent neuroinflammation [14]. Specifically, in cerebellar tissue samples, aberrant microglia and astrocyte activation was detected in combination with a massive monocyte and macrophage accumulation, particularly in the granular layer and white matter [14]. These findings correlated with marked histological abnormalities including decreased numbers of PCs and GrCs together with reduced numbers of axons within the PC layer [10,14]. Moreover, in ASD patients, increased levels of many cytokines and chemokines were described in the brain and cerebrospinal fluid, precisely, interleukin (IL)-6, tumor necrosis factor alpha (TNF-α), transforming growth factor beta 1 (TGFβ1), and C-C motif ligand 2 and 17 (CCL2 and CCL17) in the cerebellum [14,209,210,211]. Furthermore, antibodies against cerebellar proteins have also been characterized in ASD individuals and are strongly associated with impairments in behaviors, in particular deficits in social interactions and communication [212,213,214,215]. The antigenic target of these antibodies has not yet been precisely identified but robust specific reactivity was shown against cerebellar GABAergic interneurons, including Golgi cells [213,216]. Whether these antibodies alter activity of its target neurons or mark them for destruction by phagocytes requires further investigation. Remarkably, Black and Tan BRachyury (BTBR) inbred mice were identified only fifteen years ago as showing strong and consistent autism-relevant behaviors, including reduced social interactions, impaired play, low exploratory activity, unusual vocalizations and high anxiety [217,218]. These mice show a number of immunological abnormalities, many of which were described in postmortem brains of ASD subjects [219,220,221,222]. They are characterized by elevated cytokine levels in the brain, and an increased proportion of microglial cells. In particular, among the brain regions that Heo and colleagues [219] examined, the cerebellum exhibited significantly higher expression of IL-33, IL-18 and IL-6 in BTBR mice than in control, suggesting that it could be a crucial area for neuroinflammation in humans with ASD. Finally, a recent study revealed an abnormal cerebellar development (enhanced foliation and PC hypotrophy with altered dendritic spine formation) concomitant with the progression of motor impairments in BTBR mice [223]. In summary, although there is a growing body of evidence supporting the relationship between cytokine alterations and ASD, systematic and large scale investigations are needed to better clarify the role of cerebellar inflammation on the emergence of ASD and the contribution to its etiological heterogeneity.

## 4. Cerebellar Monogenic Mouse Models of ASD

Mice are the most commonly used animal model to investigate human diseases, including ASD, because they offer advantages that few species can match. Firstly, the mouse genome can be easily manipulated. Secondly, mice display similarities to humans in terms of anatomy, physiology, and genetics. For instance, the average similarity of genes between humans and mice in only protein-coding genes is 85% [224]. Thirdly, their rapid reproduction and development allow reliable and repeatable experiments to be performed at a relatively low cost. Several assays have been developed and validated to test mice for phenotype relative to ASD, including both core and associated ASD-like features [225]. Much of what we know about the pathological mechanisms underlying various disorders of the autistic spectrum comes from detailed analysis of mouse models with targeted deletions or mutations of selected candidate genes [226]. In particular, monogenic mouse models of ASD showing cerebellar alterations will be grouped as follows: those involving genes known for their role in cerebellar neurodevelopment, syndromic, and non-syndromic models. The findings are summarized in Table 1. Based on clinical parameters, ASD is generally classified as syndromic and non-syndromic. In syndromic ASD, a genetic cause is clearly defined. Non-syndromic ASD refers to the “classic” or idiopathic autism, where no distinct phenotype is present. Nonetheless, a genetic component of non-syndromic ASD has become more evident, leading to the generation of mutant mouse models to investigate these forms of ASD. Notice that mice with deficits in SHANK3 and IB2 gene expression are classified under the broad category of non-syndromic ASD models for the scope of this review. It should be noted that these genes are often co-deleted in the Phelan McDermid syndrome, which is a syndromic form of ASD. For this reason, these mouse models might as well be considered as part of the syndromic forms. Therefore, the classification proposed is arbitrary, due to the lack of a widely accepted definition in the literature [227,228].

### 4.1. Models Involving Cerebellar Development

Several genes that contribute to normal cerebellar development are consistently associated with increased susceptibility to autism. These genes include the Engrailed homeobox 2 (EN2), retinoic acid-related orphan receptor alpha (RORα), forkhead box protein 2 (FOXP2), RELN, mesenchymal-epithelial transition (MET) receptor tyrosine kinase, oncosuppressor phosphatase and tensin homolog (PTEN), Ca^2+^-dependent activator protein for secretion 2 (CADPS2), and GABRB3. The EN2 gene is required to specifically regulate mesencephalic and cerebellar development [229,230]. The RORα is a gene that plays an essential role in PC differentiation [231,232]. The FOXP2 gene promotes neuronal development, synaptic plasticity, and axon outgrowth [233]. The RELN gene, which encodes for reelin, regulates neuronal migration in the cortex and cerebellum [234]. The MET gene has been found to contribute to cerebellar growth and development, especially promoting GrC survival, differentiation, and proliferation [235,236]. The PTEN gene is involved in cell cycle control, apoptosis, and migration signaling [237]. CADPS2 contributes to normal cerebellar development by enhancing the release of brain-derived neurotrophic factor (BDNF) and neurotrophin-3 (NT-3) [238,239]. The GABRB3 gene allows for proliferation, migration, and differentiation of cerebellar precursor cells [240,241]. Genetic variants of the transcription factor Engrailed homeobox 2 (EN2) were found in 167 families associated with ASD [242]. Remarkably, knockout mice for EN2 (EN2-KO) and ASD patients exhibit similar cerebellar morphological abnormalities, including foliation patterning, hypoplasia, and decreased PC numerosity [230,243]. Furthermore, EN2-KO mice display ASD-like behaviors, including reduced sociability, impaired spatial learning/memory, and increased seizure susceptibility [244,245]. Additionally, defective motor coordination and grip strength reflexes were reported.

Retinoid-related orphan alpha receptor (RORα), specifically expressed in PCs [231,232], is another transcription factor that recently has been associated with ASD [246,247,248]. RORα knockout mice (RORα-KO; staggerer phenotype) develop a progressive PC and GrC loss (about 80%; [249]) combined with severe ataxia [250,251].

The Forkhead box protein 2 (FOXP2) mutations have been implicated in ASD and language disorders [252,253,254]. Cerebellar abnormalities were reported in mice with a disruption in the FOXP2 gene (FOXP2-KO), resulting in hypoplasia with PC migration and maturation particularly affected. Moreover, FOXP2-KO mice show impaired synaptic plasticity in PCs and motor-skill learning [255,256,257].

The RELN gene encodes for reelin, a large extracellular matrix glycoprotein. Postmortem analysis revealed that reelin expression is reduced in the cerebellum of ASD young adults [258]. In addition, mice homozygous for the reelin mutation (reeler phenotype) exhibit ataxia [259], and histological examination of their cerebella showed marked hypoplasia, aberrant PC positioning, and reduction in GrC number [260].

The MET gene (proto-oncogene receptor, tyrosine kinase), known to regulate immune function [261], is also considered the immune gene most closely associated with ASD [262,263,264,265]. MET mutant mice exhibit cerebellar hypoplasia associated with foliation defects and reduced GrC proliferation. Furthermore, altered control of balance and complex movements were reported [235].

The oncosuppressor phosphatase and tensin homolog (PTEN) stimulates cell cycle and survival by regulating phosphatidylinositol 3,4,5,-trisphosphate and Akt/protein kinase B signaling pathway [266]. Recent studies have indicated that up to 20% of children with ASD and macrocephaly presented PTEN mutations [267,268,269], suggesting that PTEN is a high-risk factor for ASD. A recent mouse model with predominantly cytoplasmic localization of PTEN, resulting in a phenotype of extreme macrocephaly and autistic-like behaviors shows an enhanced phagocytic capacity in microglia, thus indicating an aberrant microglia activation [270]. Since PTEN deletion is embryonically lethal, mice with brain-region-specific PTEN knock-out (PTEN-KO) have been investigated. A cerebellum-specific PTEN deletion led to deficits in motor coordination and moderate-to-severe seizures associated with cellular hypertrophy, particularly of GrCs, without evidence of abnormal proliferation [271]. Additionally, these changes were accompanied by modifications in different protein levels, including A-type potassium channel Kv4.2, NR2A subunit, and mGluR1/5 [272]. When PTEN deletion is induced specifically in PCs (PTEN-KO PC), these neurons appear hypertrophic with thicker dendrites and axons. Functional analysis revealed that PTEN-KO PC showed a reduced firing rate and increased amplitude of EPSC evoked by stimulation of PF-PC synapse [273]. Finally, loss of PTEN in PCs determines ASD-like traits, such as impaired sociability and repetitive behavior [273].

Newly identified developmentally regulated genes in the cerebellum include Ca^2+^-dependent activator protein for secretion 2 (CADPS2), which is specifically expressed in cerebellar GrCs [238], where it promotes brain-derived neurotrophic factor (BDNF) and neurotrophin-3 (NT-3) release [239,274]. Several studies have found rare variants in the CADPS2 gene associated with ASD [275,276,277]. Mice deficient in CADPS2 (CADPS2-KO) display both cerebellar morphological and functional abnormalities. Delayed development, vermis hypoplasia, increased GrC apoptosis, and reduced PC number accompanied by aberrant dendritic arborization were observed [239,277]. Furthermore, EM analysis of PF-PC synapses detected, especially in lobules VI and VII, an enlargement of presynaptic boutons resulting in paired-pulse facilitation impairment [239,277]. Finally, CADPS2-KO mice exhibit ASD-like behavioral phenotypes, including impaired sociability, hyperactivity in a familiar environment, decreased exploratory behavior, and increased anxiety in a novel environment [239,277].

Cook and colleagues [278] found a cluster of GABAA receptor subunit genes containing GABAA receptor β3 (GABRB3), α5 (GABRA5), and γ3 (GABRG3) within Angelman Syndrome Chromosome Region (15q11–q13). Several genome-wide association studies have indicated the GABRB3 gene as an excellent candidate for ASD 279–281]. Remarkably, GABRB3 expression was found to be reduced in the parietal cortex and cerebellum of ASD individuals [279]. GABRB3-KO mice exhibit impaired social and exploratory behaviors and cerebellar hypoplasia, especially in vermis lobules [240]. In addition, these mice are hyperactive and perform motor-skill tasks poorly [240,279].

### 4.2. Models of Syndromic ASD

Several human syndromes caused by a mutation in one gene or rare genomic copy number variation (CNV) increase the risk of ASD diagnosis. The monogenic syndromes, in which the cerebellum has been consistently implicated, are Fragile X syndrome with a mutation in FMR1 [280], Rett syndrome with a mutation in MECP2 [281], and tuberous sclerosis with mutations in TSC1 or TSC2 [282]. Finally, deletions or duplications that occur at the 15q11–q13 chromosome results in Angelman and Dup15q syndromes [283].

Fragile X syndrome (FXS) is characterized by intellectual disability and is the most common monogenic cause of ASD, accounting for approximately 5% of all cases (for review, see [284,285]). Patients with FXS exhibit ASD-like traits, including perseverative behaviors and cognitive inflexibility, abnormal sensory responses, and social and communication impairments [286]. FXS occurs as a result of FMR1 gene mutation, which leads to loss of fragile X mental retardation protein expression (FMRP) [264,265]. Structural MRI studies have described cerebellar volume alterations associated with PC loss in FSX subjects [287,288,289,290,291]. Furthermore, cerebellum-dependent eyeblink conditioning, a simple form of associative learning, is significantly impaired in these patients [292,293]. Recently, defects in immune cells have emerged as potential critical contributors to FXS pathophysiology. Postmortem analyses of FXS patients have described an aberrant differentiation in human neural progenitor cells and, in particular, a significant induction of the astrocyte marker glial fibrillary acidic protein (GFAP) [294]. In parallel, some studies also showed marked astrocyte activation and cytokine imbalance with increased IL-6 levels in the FMR1-KO mouse cerebellum [295,296], indicating a possible increase in inflammation on both human and mouse models. Furthermore, FMR1-KO mice present several behavioral deficits, including an attenuate eyeblink conditioning [292], and cerebellar abnormalities similar to those observed in ASD patients [297,298]. Specifically, the most prominent changes observed were reduced volume and DCN cell loss [299], elongated dendritic spines on PCs, and enhanced LTD induction at PF-PC synapses [292]. Further evidence for the potential role of cerebellar dysfunction in FXS includes deficient cerebellar-induced dopamine release on the mPFC in these mice [158].

Rett syndrome (RTT) is a progressive neurodevelopmental disorder that manifests mainly in females leading to language and motor impairments, ASD behavior and severe intellectual disability [300,301,302,303]. It is caused by mutations in the X-linked MECP2 gene, which encodes methyl-CpG-binding protein 2 [304]. MeCP2 likely plays an essential role in regulating different sets of genes relevant to the RTT pathogenesis, including brain-derived neurotrophic factor BDNF [305,306,307]. Interestingly, several studies demonstrated that the immune system is involved in RTT in early life, particularly microglia activation [308]. Maezawa and colleagues suggested that RTT microglia are sensitive to both immunological stimuli and neuronal/astrocytic signals causing neuroinflammation and, consequently, affecting brain development [309,310]. Postmortem studies of the cerebella of patients with RTT have consistently revealed atrophy, loss of PCs, Bergmann gliosis, and loss of myelin in the white matter [311,312]. MeCP2-KO mice, which recapitulate the gross anatomical abnormalities of the human phenotype, display a reduced cerebellar volume largely due to smaller and denser packing of GrCs [313,314]. In addition, these animals display deficits in motor learning accompanied by irregular PC firing [315]. The RTT-like features observed in MeCP2-KO mice are linked to decreased cerebellar BDNF protein levels [316]. Intriguingly, a postnatal BDNF overexpression in the brain leads to locomotor and electrophysiological improvements [317]. Accordingly, BDNF levels are shown to be decreased in postmortem brain tissues from human patients, suggesting that BDNF could be an alternative therapeutic for RTT [318,319,320].

Tuberous sclerosis complex (TSC) is a multisystem autosomal dominant disorder characterized by the presence of malformed tissues (tubers) or hamartomas (benign tumors) in multiple organs, including the brain, and is often associated with a wide range of cognitive, behavioral, and psychiatric manifestations [321,322]. For instance, an estimated 40–50% of individuals with TSC develop ASD [323,324,325]. Imaging studies and postmortem examinations have reported the presence of cerebellar tubers in TSC patients associated with focal atrophy, extensive degeneration of PCs, and reactive astrogliosis [326,327,328,329,330]. Noticeably, a correlation between cerebellar tubers and ASD was first reported in 2000 [331,332]. TSC is caused by mutations in either TSC1 or TSC2 genes, which encode hamartin and tuberin proteins, respectively. These proteins form a tumor suppressor complex that negatively regulates cell growth and proliferation through rapamycin (mTOR) signaling [333,334,335,336]. Of note, morphological and functional changes in glial cells involving astrocytes, oligodendrocytes, microglia, and activation of inflammatory signaling pathways are histopathological characteristics of TSC [337,338,339,340]. In the mouse cerebellum in vivo, tuberin is primarily restricted to the perinuclear region of PCs, while hamartin is localized in neuronal or astrocytic processes [341]. The impairment of social, repetitive, and communicative responses that mimics the human ASD condition has been reported in a mouse model that presents a PC-specific deletion in TSC1 [342]. At the cellular level, these behavioral deficits are associated with reduced excitability, abnormal spine density, and PC loss [342]. Notably, the reduction of PC firing rate is correlated with ataxia, motor deficits, and abnormalities in eyeblink conditioning [342,343]. Similar findings were reported in PC-specific TSC2-KO mice [344]. Interestingly, the mTOR inhibitor rapamycin rescued PC-specific alterations and ameliorated behavioral deficits in both TSC1 and TSC2 KO mice [342,345].

Angelman syndrome (AS) is a neurodevelopmental disorder characterized by delayed development, absent speech, intellectual disability, movement and balance impairments, and sometimes seizures (for review, see [344,346]). It is caused by loss-of-function mutations of the UBE3A gene located within a region of chromosome 15 known as 15q11–q13, which encodes ubiquitin protein ligase E3A [347]. UBE3A gene has been also linked to ASD [348,349,350]. Previous studies have quantified that about 50% of AS individuals show signs and symptoms of ASD [351,352]. Indeed, overexpression of the UBE3A gene is among the more common source of genetic risk factors for ASD [353,354]. In both humans and mice, the UBE3A gene has been reported to be transcribed preferentially from the maternal alleles in specific brain regions, including the hippocampus, olfactory bulb, and cerebellum [355,356,357]. UBE3A maternal-deficient mice display UBE3A expression preferentially in PCs without changes in gross cerebellar morphology [358,359]. Behavioral tests for motor coordination have revealed ataxic gait, coordination and balance deficits [358,359]. Finally, using in vivo electrophysiology, fast oscillation (160 Hz) in the cerebellar cortex sustained by abnormally increased PC firing rate was found [360]. In a recent study, Egawa and colleagues [361] suggested that the cerebellar-related movement defects in UBE3A maternal-deficient mice might not only be due to PC dysfunction but also to the upregulation of GABA transporter 1 (GAT1), resulting in reduced GABA concentrations in the extrasynaptic space and thus decreased tonic inhibition of GrCs. Interestingly, pharmacological compensation of reduced tonic inhibition by delivering a selective GABAA receptor agonist ameliorated motor impairments in these mice [361].

The dup15q syndrome is a neurodevelopmental disorder caused by various duplications of the region of the 15q11–q13 chromosome. It is characterized by hypotonia resulting in gross and fine motor delays, cognitive impairments, and seizures [362,363,364]. The dup15q syndrome is the most prevalent chromosomal anomaly associated with ASD, occurring in approximately 1–3% of cases [350,365,366]. The mouse model for the human paternally inherited 15q11–13 duplication (patDp/+) exhibits a reduction in social interaction together with emission of few ultrasonic vocalizations, behavioral inflexibility, and motor coordination and learning (eyeblink conditioning) deficits [367,368]. Moreover, cerebellar LTD at PF-PC synapses and the normal regression of multiple innervations of PCs by CFs were found to be impaired [367].

### 4.3. Models of Non-Syndromic ASD

Other genes displayed higher expression in the cerebellum, such as SHANK1-3 or NLGN3, which have been implicated recently in the pathogenesis of non-syndromic ASDs [369,370,371,372,373]. Furthermore, deletion of SHANK3, a distal gene of chromosome 22, results in 22q13.3 deletion syndrome, also called Phelan–McDermid syndrome (PMS; [374]), which has been associated with cases of ASD at a rate of about 0.5–2% [375]. Interestingly, in ASD individuals, several studies have found SHANK3 to be disrupted by deletions ranging from hundreds of Kb to Mb, causing the co-deletion of the IB2 gene [370,376,377,378]. Herein, together with reporting mutant mice for SHANK1-3 and NLGN3, we will describe a new murine model lacking the IB2 gene characterized by abnormalities in cerebellar anatomy and deficits in motor, social, and cognitive tasks, showing an ASD-like phenotype [369,370].

SHANK1, SHANK2, and SHANK3 constitute a family of scaffolding proteins that are part of the postsynaptic density (PSD) in glutamatergic synapses and link receptors to the actin cytoskeleton [379]. SHANK proteins have been implicated in spinogenesis and synapse development, maturation, and stability [379,380]. SHANK family proteins are found in diverse regions of the murine brain, including the cerebellum [381,382,383,384]. SHANK1 and SHANK2 are more abundant in PCs and their dendrites, while SHANK3 was expressed only in GrCs [385]. To better understand the contribution of each SHANK protein in ASD, mouse models have been generated. The first model was created by deleting SHANK1 [386] and resulted in ASD-like traits, such as repetitive behaviors, impaired ultrasonic vocalizations, and motor coordination with balance deficits [387,388,389]. Recently, two studies have independently reported cerebellar functions of SHANK2, which used mice carrying the deletion of SHANK2 exons 6/7 (SHANK2 e6/7-KO) or exon 7 (SHANK2 e7-KO), and specifically in PCs (PC(Pcp2)-specific SHANK2-KO or PC(L7)-specific SHANK2-KO) [390,391]. Notably, these murine models exhibit diverse ASD-like behaviors and distinct cellular phenotypes. Both PC(L7)-specific SHANK2-KO and SHANK2 e6/7-KO mice exhibit motor coordination and learning impairments, social interaction deficits, altered ultrasonic vocalizations, repetitive behavior, and hyperactivity [390,391]. Conversely, PC(Pcp2)-specific SHANK2-KO display motor coordination but not ASD-like behaviors [391]. In SHANK2 e7-KO mice, electrophysiological recordings unveiled impaired plasticity at the PF-PC synapses, increased inhibitory inputs onto PCs and a significant increase in the irregularity of PC firing, selectively occurring in posterior (but not anterior) lobules [390]. The difference in the excitability of anterior (I-V) vs. posterior (X) lobules observed in global SHANK2-KO mice is quite intriguing since posterior abnormalities (i.e., a decrease in grey matter) have been described in ASD patients [392]. In contrast, the deletion of SHANK2 exons 6/7 did not affect the plasticity at PF-PC synapses [391]. These differences might be attributable to SHANK2 KO mice carrying a different exon deletion and/or to different time courses of Pcp2/L7 and SHANK2 expressions [391]. Multiple isoform lines of SHANK 3 mutant mice, due to the transcriptional complexity of this gene, have been described and several display ASD-like behaviors to various degrees [393,394,395,396]. Although each of these models has some construct validity, only one represents an accurate mimic of a human mutation of SHANK3. This mutation affects exon 21, which encodes a truncated SHANK3 protein lacking the C-terminal region [376,397]. Mice with C-terminal deleted SHANK3 (SHANK3-ΔC) exhibit social abnormalities, repetitive behaviors, novelty avoidance, and cerebellar deficits which comprise impaired motor coordination and learning [343,398,399]. Moreover, SHANK3-ΔC mice showed a decreased number of PCs, often with fewer dendritic spines [343,400].

The neuroligin (NLGN) family of postsynaptic cell adhesion molecules are involved in synapse formation and maturation by interacting with presynaptic neurexins. Mutations in the X-linked NLGN3 gene, which encodes for neuroligin-3 (NLGN3), have been associated with ASD phenotype [401]. Two mouse models have been generated to study NLGN3 dysfunction: NLGN3-KO mice that present a complete loss of NLGN3 protein expression [402], and knock-in mice expressing the NLGN3 arginine to cysteine point mutation (NLGN3-R451C) that recapitulate the same mutation described in two brothers with ASD [403]. Both models display abnormal social and repetitive behaviors [404,405,406]. NLGN3-R451C mutant mice showed a marked reduction of NLGN3 protein expression in the cerebellum. The elimination of redundant CF to PC synapses was transiently impaired, becoming normal after two weeks of age. In addition, mIPSC frequency in PCs was enhanced, and somatic calcium transients induced by CF multi-innervation inputs were reduced [407]. Conversely, NLGN3-KO mice exhibit a weak phenotype in inhibitory synaptic transmission [408]. A few studies have investigated the role of NLGN3 protein on long-term synaptic plasticity. Specifically in the cerebellum, Baudouin and colleagues reported that NLGN3-KO mice showed motor coordination deficits accompanied by loss of mGluR-mediated LTD at PF-PC synapses [409].

Deletions in chromosome 22q13.3 are correlated with some types of ASDs and with the Phelan–McDermid syndrome (PMS) in humans [374,375,410]. The deletions extend proximally from the SHANK3 gene by at least 0.8 Mb, thus co-deleting the IB2 gene in almost all the documented PMS cases. This gene is also known as MAPKIP2 or JIP2 and is located 70 kbp from SHANK3 [370,376]. The Islet Brain-2 protein (IB2) is expressed in both neurons and neuroendocrine cells [411] and is largely enriched in the PSDs of the cerebral cortex and cerebellar glomeruli [370], where it can serve as a scaffolding protein that regulates p38 MAP kinase signaling downstream of NMDAR activation [412]. Given that SHANK3 mutation alone could not account for all the phenotypes observed in patients with PMS [413,414,415], Giza and colleagues developed a mutant mouse model deficient in the IB2 gene to investigate its specific role in cerebellar functions [370]. IB2-KO mice display morphologically normal PSDs with unaltered molecular composition (save IB2 deficiency) but show enhanced NMDAR-mediated glutamatergic transmission at the MF-GrC synapse, altered PC morphology, motor deficits (poor performance on an accelerating rotarod), and cognitive deficits (reduced exploratory behavior and social interaction) [370]. Therefore, IB2-KO mice associate cerebellar impairment and ASD-like symptoms, which is entirely consistent with the cerebellar phenotype of PMS patients, further supporting the cerebellar role in ASDs. In particular, granular layer activity was found altered at different levels. A remarkable increase in intrinsic GrC excitability was likely due to the tonic activation of AMPARs/NMDARs by glutamate and deregulation of voltage-gated Na^+^ and K^+^ currents (A-type, delayed rectifier, and Ca^2+^-activated) was observed [369]. Moreover, an alteration in the E/I balance was described at the cellular (increased NMDARs-mediated currents with normal inhibitory postsynaptic currents in GrCs) and spatial levels in the granular layer [369]. Indeed, by using voltage-sensitive dye imaging (VSDi), an altered spatial organization of granular layer activity was observed in IB2-KO mice, with a shift from a classic “Mexican hat” to a “stovepipe hat” profile, characterized by enhanced excitation cores and poor inhibitory surrounds [369] (Figure 5). Moreover, whole-cell patch-clamp recordings revealed enhanced LTP at the MF-GrC synapse (probably due to increased quantum size and quantum content of presynaptic vesicles) in KO mice, whereas VSDi revealed an increase in LTP/LTD area, mirroring the altered E/I center-surround structure [369]. Since NMDAR expression in the granular layer remarkably overwhelms that in the molecular layer and DCN [416], IB2-KO-dependent NMDAR hyperfunctioning and center-surround alterations might be crucial to ASD pathogenesis. Accordingly, massive and selective hyperfunctioning of NMDARs in neocortical microcircuits has been proposed to lead to hyper-memory, hyper-attention, and hyper-perception (the so-called Intense World Theory), whereas microcircuit glutamatergic hyper-functionality in the amygdala could result in hyper-emotionality [417]. Rinaldi and colleagues showed that enhanced NMDARs-mediated glutamatergic neurotransmission resulted in enhanced E/I balance that was correlated to hyper-reactivity and hyperplasticity in the valproate acid model of ASD [418]. Specific studies reported that the cortical hyperconnectivity was confined to the minicolumnar range [418]. The minicolumn has been proposed to be the basic processing unit of the mature cortex [419,420,421,422], whose alterations have been implicated in ASD [423]. Interestingly, the shift from the Mexican hat to the stovepipe hat profile, reported in the cerebellar cortex of the IB2-KO mouse, has been hypothesized by Casanova and colleagues on the basis of anatomical alterations in neocortical minicolumns in postmortem ASD human brains [423,424], thus altering receptor and cognitive fields in a way that might explain the cognitive dysfunction of ASD [423]. It is intriguing to speculate that the alterations found in the IB2 KO mice strongly resemble those reported for the neocortex and associative brain areas (altered E/I balance, plasticity, and spatial organization of activity, Figure 5), supporting the already considerable evidence that cerebellum alterations may be crucial to ASD pathogenesis.

## 5. Conclusions

Cerebellar involvement in ASD pathophysiology is now an acquired knowledge. The cerebellum is connected to the main areas at the core of the disease (as the PFC, amygdala, hippocampus, and the social brain in general) and shows alterations in its structure, function, and connectivity which might impact on ASD pathophysiology and, ultimately, on ASD phenotype. Nevertheless, the heterogeneity of these disorders and the extension of the brain areas involved (from the neocortex to subcortical areas and the cerebellum) make it extremely difficult to precisely identify the cerebellar alterations and their impact on the rest of the brain (and on the phenotype). Therefore, the use of murine models to investigate the genetic component of the disease and, beyond that, the impact of alterations in different brain areas on the autistic phenotype is unavoidable. To date, it seems that anatomical and morphological alterations in the cerebellum are a common trait in autistic patients as well as in murine ASD models. These alterations are accompanied by neurophysiological impairments at the cellular and network levels, usually resulting in altered E/I balance, which is likely to impact the spatiotemporal properties of neuronal network processing. Moreover, immune dysfunction seems to play a key role in ASD pathogenesis and the nature of neuroinflammation mechanisms in ASD deserves further investigation.

## Figures and Tables

**Figure 1 ijms-23-03894-f001:**
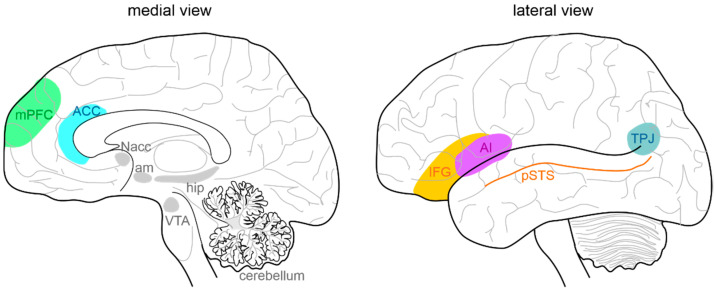
Anatomy of the social brain. The main brain areas involved in the “social brain” are reported in the medial (**left**) and lateral (**right**) schematic view of the human brain: medial prefrontal cortex (mPFC), anterior cingulate cortex (ACC), temporo-parietal junction (TPJ), posterior superior temporal sulcus (pSTS), inferior frontal gyrus (IFG), and anterior insula (AI). The main regions connected to the “social brain” are reported in grey: hippocampus (hip), amygdala (am), ventral tegmental area (VTA), nucleus accumbens (NAcc), and cerebellum.

**Figure 2 ijms-23-03894-f002:**
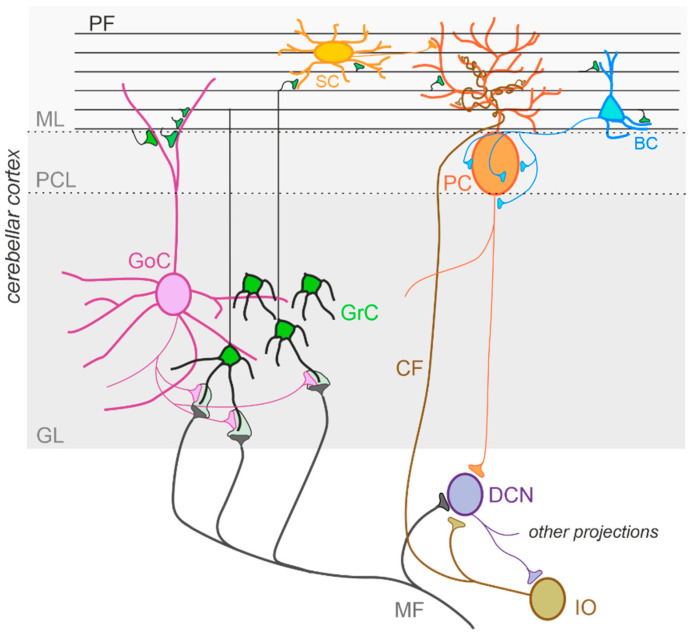
The cerebellar circuit. Schematic representation of the main components of the cerebellar circuit. The inputs are provided by mossy fibers (MF) and climbing fibers (CF), these latter originating in the inferior olive (IO). Both inputs send collaterals to the deep cerebellar nuclei (DCN) before entering the cerebellar cortex. Granule cells (GrC) and Golgi cells (GoC) are the main neuronal types present in the granular layer (GL) of the cerebellar cortex. GrC axons reach the molecular layer (ML) where they bifurcate originating the parallel fibers (PF). The inhibitory interneurons in the ML are stellate cells (SC) and basket cells (BC), which inhibit Purkinje cells (PC) in the Purkinje cell layer (PCL). The PC provides the output of the cerebellar cortex, inhibiting DCN neurons, which in turn provide the main output of the cerebellar circuit. Notice that DCN project to the IO, generating a loop mediated by the CF.

**Figure 3 ijms-23-03894-f003:**
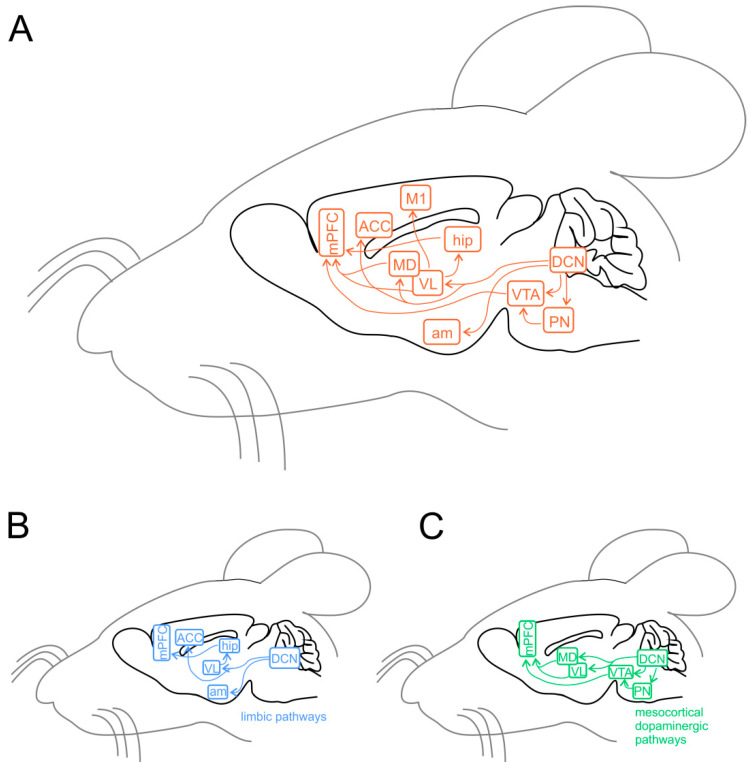
Cerebellar connectivity to other brain areas. The cerebellum is one of the most interconnected structures in the brain. (**A**) Schematic representation of the mouse brain and the main cerebellar connections thought to be relevant for its role in ASD. DCN, deep cerebellar nuclei; PN, pontine nuclei (including reticulo-tegmental nuclei and pedunculopontine nuclei); VTA, ventral tegmental area; am, amygdala; hip, hippocampus; VL, ventrolateral thalamic nucleus; MD, mediodorsal thalamic nucleus; M1, primary motor cortex; ACC, anterior cingulate cortex; mPFC, medial prefrontal cortex. (**B**) Same representation as in (**A**), showing the pathways involving the limbic system. (**C**) Same representation as in (**A**), showing the connections involved in the mesocortical dopaminergic pathways, regulating mPFC activity modulation.

**Figure 4 ijms-23-03894-f004:**
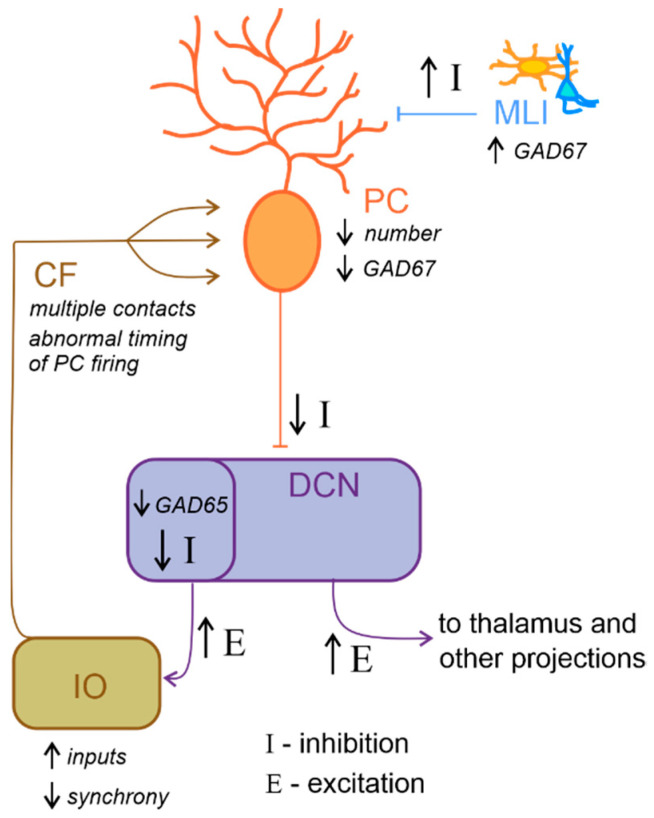
Altered excitatory/inhibitory balance in the cerebellum-inferior olive loop in ASD. Schematic representation of the main alterations described in the cerebellum-inferior olive circuit in ASD, as described in the main text. Briefly, Purkinje cells (PC) are reduced in number and show a decreased level of GAD67 mRNA expression, while molecular layer interneurons (MLI) show an increased inhibition over PC. These would likely determine a decrease in the inhibition (I) over deep cerebellar nuclei (DCN) neurons. DCN disinhibition would increase the excitatory (E) level increasing the output towards the thalamus and other brain regions. Concerning the loop with the inferior olive (IO), DCN neurons which project to this area show decreased GAD65 mRNA expression levels, thus resulting in a decreased inhibition over IO neurons, likely increasing the excitatory inputs and decreasing synchronicity. In ASD, multiple climbing fibers (CF) impinging on the same PC were described. Together with the alterations in IO activity, this anatomical abnormality likely contributes to impair the timing of PC spiking activity.

**Figure 5 ijms-23-03894-f005:**
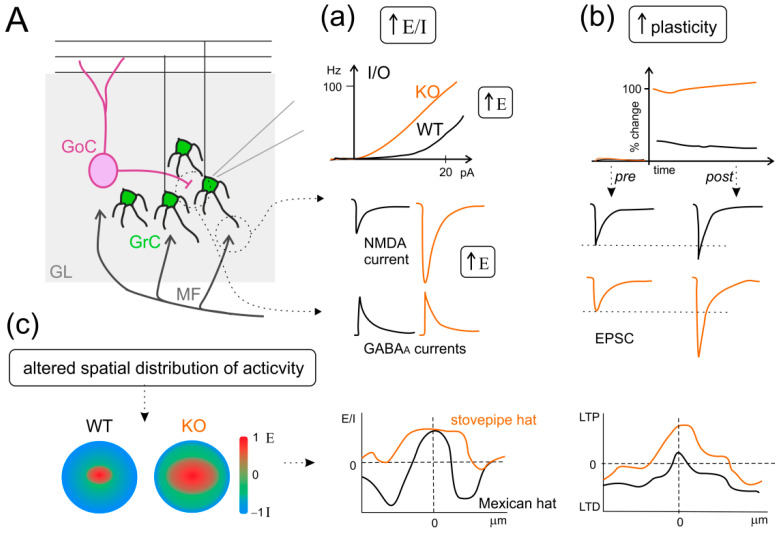
The IB2 KO mouse model as an example of increased E/I balance, hyper-plasticity, and altered spatial organization of activity in ASD. (**A**) Schematic view of the granular layer (GL) microcircuit, with mossy fibers (MF) inputs, granule cells (GrC), and Golgi cells (GoC). Three main panels describe the alterations observed in the IB2 KO mouse model of ASD. (**a**) Increased excitatory/inhibitory (E/I) balance: the additional panels show the input-output (I/O) relationship in granule cells, the NMDA component of excitatory postsynaptic currents in response to MF stimulation, and inhibitory postsynaptic currents, in both WT (black) and KO (orange) conditions. (**b**) Enhanced long-term potentiation (LTP): the additional panels show the time-course of excitatory postsynaptic currents (EPSC) percent change before and after LTP induction, and EPSC traces pre- and post-induction, for both WT and KO conditions, as in (**a**). (**c**) Altered spatial distribution of activity in the granular layer: the additional panels show the “classic” organization in center/surround structures, with excitation prevailing in the core and inhibition in the surrounds. This organization shifts from the Mexican hat to the stovepipe hat profile. Interestingly, this alteration is preserved after plastic changes in synaptic activity, where LTP and long-term depression (LTD) organize mirroring the E/I profile. ((**a**–**c**) panels are drawn from the results shown in [370]).

**Table 1 ijms-23-03894-t001:** This table summarizes the main abnormalities reported in the cerebellum in mouse models of ASD.

Mouse Model	Neurochemical Changes	Structural/Cellular Abnormalities	Functional Abnormalities	Behavioral Deficits
EN2-KO		■ Foliation ■ Hypoplasia ■ ↓ PC number	■ Motor coordination ■ Grip strength reflexes ■ ↑ Seizure susceptibility	■ ↓ Sociability ■ ↓ Spatial memory
RORα-KO (*staggerer*)		■ ↓ PC number ■ ↓ GrC number	■ Ataxia	
FOXP2-KO		■ Hypoplasia ■ PC migration ■ PC maturation	■ PC synaptic plasticity ■ Motor learning	■ ↓ Vocalization
Reelin-mutant (*reeler*)		■ Hypoplasia ■ PC positioning ■ ↓ GrC number	■ Ataxia	
MET-mutant		■ Foliation ■ Hypoplasia ■ ↓ GrC proliferation	■ Balance control ■ Complex movements	
PTEN-KO (*cerebellum*)	■ ↑ Kv4.2 ■ ↑ NR2A subunit ■ ↑ mGluR1/5	■ Hypertrophy ■ ↑ GrC soma size	■ Motor coordination ■ ↑ Seizure susceptibility	
PTEN-KO (*Purkinje cell*)		■ Enlarged soma ■ Thicker axons and dendrites	■ ↓ PC firing rate ■ ↓ EPSC amplitude (PF-PC synapse)	■ ↓ Sociability ■ Repetitive behavior
CAPDS2-KO		■ Vermis hypoplasia ■ ↑ GrC apoptosis ■ ↓ PC number ■ Aberrant PC arborization ■ Enlarged PF terminal boutons	■ ↓ PPF at PF-PC synapse	■ ↓ Sociability ■ Hyperactivity ■ ↓ Exploratory behavior ■ ↑Anxiety
GABRB3-KO		■ Vermis hypoplasia	■ Motor coordination	■ ↓ Sociability ■ Hyperactivity ■ ↓ Exploratory behavior
FMR1-KO		■ Hypoplasia ■ ↓ DCN cell number ■ Enlogated PC spines	■ ↓ Eye-blink conditioning ■ ↑ LTD at PF-PC synapse ■ ↓ cerebellar-induced dopamine release on mPFC	■ Repetitive behavior ■ Hyperactivity
MeCP2-KO	■ ↓ BDNF	■ Hypoplasia ■ ↓GrC soma size ■ ↑ densely packed GrC	■ Motor learning ■ Irregular PC firing	↓
TSC1-KO TSC2-KO (*Purkinje cell*)		■ ↓ PC number ■ Abnormal PC spine density	■ ↓ Eye-blink conditioning ■ ↓ PC firing rate ■ Ataxia	■ ↓ Sociability ■ Repetitive behavior
UBE3A-KO	■ GAT1 upregulation		■ Motor coordination ■ Balance control ■ ↑ PC firing rate ■ Ataxia	
patDp/+		■ Multiple innervation of PCs by CFs	■ Motor coordination ■ ↓ Eye-blink conditioning	■ ↓ Sociability ■ ↓ Vocalization ■ Behavioral inflexibility
SHANK1-KO			■ Motor coordination ■ Balance control	■ Repetitive behavior ■ ↓ Vocalization
SHANK2 e6/7-KO			■ Motor coordination	■ ↓ Sociability ■ Repetitive behavior ■ ↓ Vocalization ■ Hyperactivity
SHANK2 e7-KO			■ ↓ LTD at PF-PC synapse ■ Irregular PC firing ■ ↑ Inhibition inputs onto PCs	
SHANK2-KO (L7 *Purkinje cell*)				■ ↓ Sociability ■ Repetitive behavior ■ ↓ Vocalization ■ Hyperactivity
SHANK2-KO (Pcp2 *Purkinje cell*)			■ Motor coordination	
SHANK3-ΔC		■ ↓ PC number ■ ↓ PC spine density	■ Motor coordination ■ Motor learning	■ ↓ Sociability ■ Repetitive behavior ■ Novelty avoidance
NLGN3-KO			■ Motor coordination ■ ↓ LTD at PF-PC synapse	■ ↓ Sociability ■ Repetitive behavior
NLGN3-R451C	■ ↓ NLGN3 protein exporession		■ ↑ mIPSC frequency ■ ↓ Ca^2+^ transient induced by CF inputs ■ Multiple innervation of PCs by CFs	■ ↓ Sociability ■ Repetitive behavior
IB2-KO		■ Thinner PC dendrites ■ Shorter PC dendritic arbor	■ Motor learning ■ ↑ GrC excitability ■ ↑ NMDA-EPSC ■ Deregulation of voltage-gated Na^+^ and K^+^- currents ■ Alteration in E/I balance ■ ↑ LTP at MF-GrC synapse	■ ↓ Sociability ■ ↓ Exploratory behavior

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
