# Peer review of "The Cerebellar Involvement in Autism Spectrum Disorders: From the Social Brain to Mouse Models"

_ijms, 2022, doi:10.3390/ijms23073894_

Round 1

Reviewer 1 Report

This a very interesting and rather complete narrative review on the role of cerebellum in the pathophysiology of autism spectrum disorder, based on updated literature data on humans and animal models.

Literature data are clearly reported and discussed. English style is adequate.

Only one comment. Please avoid to use the term "mental retardation" in the text; it has been finally substituted by "intellectual disability" in the DSM 5. 

Author Response

This a very interesting and rather complete narrative review on the role of cerebellum in the pathophysiology of autism spectrum disorder, based on updated literature data on humans and animal models.

Literature data are clearly reported and discussed. English style is adequate.

We thank the Reviewer for the positive evaluation of our manuscript.

Only one comment. Please avoid to use the term "mental retardation" in the text; it has been finally substituted by "intellectual disability" in the DSM 5. 

We thank the Reviewer for pointing this out. The text has been amended.

Reviewer 2 Report

The review written “The cerebellar involvement in autism spectrum disorders: from the social brain to mouse models”. Below are a series of comments for the authors.

  1. Abstract should be revised: In the abstract section, the authors should write the background of the research. What motivated the researcher to do this research? Why this review is important?.
  2. Authors should include a table/cartoon showing the inflammatory pathways. Diagrammatic representations which are very important in review. You should have at least 3 to 4 diagrammatic representations.
  3. Introduction is very short, authors should be providing more literature, I suggest the authors to elaborate it. The introduction needs to revision based on the update literatures. However, authors should provide additional references to show inflammatory mediators which are associated with ASD (Prog Neuropsychopharmacol Biol Psychiatry. 2017 Oct 3;79(Pt B):472-480; Eur J Pharmacol. 2018 Jun 15; 829:70-78.).
  4. It is also important to discuss how inflammatory mediator are involved in the ASD disease progression.
  5. The authors should explain the inflammatory mechanisms involved in animal and human diseases.
  6. Discussion should be revised: It is very important to mention the role of immune transcription factor signaling pathways in each section (Behav Brain Res. 2019 May 17; 364:213-224; Prog Neuropsychopharmacol Biol Psychiatry. 2019 Jan 10; 88:352-359. Prog Neuropsychopharmacol Biol Psychiatry. 2017 Oct 3;79(Pt B):184-191.).
  7. The authors should carefully re-organize the structure errors and language should be improved.

Author Response

The review written “The cerebellar involvement in autism spectrum disorders: from the social brain to mouse models”. Below are a series of comments for the authors.

We thank the Reviewer for the comments, which allowed to integrate the manuscript with the topic of neuroinflammation in ASD, which was under-represented in the initial version. We integrated Reviewer’s suggestions by citing the topic of neuroinflammation and its relevance to ASD in the Introduction, and then expanding it in a dedicated section in the main text “Cerebellar inflammation in ASD”. We also reported the related alterations seen in the mouse models, where they are known, in the second part of the review.

    1) Abstract should be revised: In the abstract section, the authors should write the background of the research. What motivated the researcher to do this research? Why this review is important?

Approaching ASD research is often complicated by the multiple aspects of these syndromes, the lack of a unitary view of the diseases, and the difficulty to find all the available information in an organized way. The scope of this review is to provide an organized view of what is known about the cerebellar involvement in ASD and the main monogenic mouse models that can be used to study cerebellar dysfunctions in these diseases. It can also be of help for other researchers (or clinicians) to get insight into cerebellar contribution to ASD, though not working directly with the cerebellum and finding then difficult to search the literature from the start. This perspective is now reported more clearly in the abstract section.

    2) Authors should include a table/cartoon showing the inflammatory pathways. Diagrammatic representations which are very important in review. You should have at least 3 to 4 diagrammatic representations.

We thank the Reviewer for this suggestion. However, we already have 5 figures and 2 of them are made in a diagrammatic-like style (Figs.3-4). Moreover, this review focuses on cerebellar role in ASD and the use of monogenic mouse models with cerebellar phenotype. We added a section on inflammation in ASD cerebellum and described a (non-monogenic) mouse model used to investigate inflammation related to ASD, using the papers the Reviewer suggested as a reference. However, it is beyond the scope of this review to comprehensively summarize the inflammatory pathways, which are very complex and deserve a review of their own. We therefore cited specialized reviews on the subject so that the interested readers can find sources on detailed discussions of the topic.

    3) Introduction is very short, authors should be providing more literature, I suggest the authors to elaborate it. The introduction needs to revision based on the update literatures. However, authors should provide additional references to show inflammatory mediators which are associated with ASD (Prog Neuropsychopharmacol Biol Psychiatry. 2017 Oct 3;79(Pt B):472-480; Eur J Pharmacol. 2018 Jun 15; 829:70-78.).

The Introduction is thought to provide a general background of the review, which is not focused on the multiple causes of ASD but on the cerebellar involvement, as far as we know. Nevertheless, we cited more than 20 papers, mostly spanning the last 15 years. We anticipated here the topic of the neuroinflammation, that is developed more in a dedicated section in the main text of the review (in the core of the description of cerebellar abnormalities in ASD).

  4)  It is also important to discuss how inflammatory mediator are involved in the ASD disease progression.

We added a section on “Cerebellar inflammation in ASD”).

 5)   The authors should explain the inflammatory mechanisms involved in animal and human diseases.

We added a section on “Cerebellar inflammation in ASD”).

  6)  Discussion should be revised: It is very important to mention the role of immune transcription factor signaling pathways in each section (Behav Brain Res. 2019 May 17; 364:213-224; Prog Neuropsychopharmacol Biol Psychiatry. 2019 Jan 10; 88:352-359. Prog Neuropsychopharmacol Biol Psychiatry. 2017 Oct 3;79(Pt B):184-191.).

Besides adding an entire section on inflammation in the ASD cerebellum, we integrated the description of mouse models where the involvement of the immune system is described. We also cited the importance of this topic in the Conclusions.

   7) The authors should carefully re-organize the structure errors and language should be improved.

The text has been amended for errors in the structure and language.

Round 2

Reviewer 2 Report

The authors have satisfactorily responded to my questions. I have no issues in accepting the manuscript.

Author Response

We thank the Reviewer for the positive evaluation of our work.